

# Forest birds respond to the spatial pattern of exurban development in the Mid-Atlantic region, USA

Marcela Suarez-Rubio[1] and Todd R. Lookingbill[2]

[1] Institute of Zoology, University of Natural Resources and Life Sciences, Vienna, Austria
[2] Department of Geography and the Environment, University of Richmond, Richmond, VA, United States

Corresponding author
Marcela Suarez-Rubio,
marcela.suarezrubio@boku.ac.at

## ABSTRACT

Housing development beyond the urban fringe (i.e., exurban development) is one of the fastest growing forms of land-use change in the United States. Exurban development's attraction to natural and recreational amenities has raised concerns for conservation and represents a potential threat to wildlife. Although forest-dependent species have been found particularly sensitive to low housing densities, it is unclear how the spatial distribution of houses affects forest birds. The aim of this study was to assess forest bird responses to changes in the spatial pattern of exurban development and also to examine species responses when forest loss and forest fragmentation were considered. We evaluated landscape composition around North American Breeding Bird Survey stops between 1986 and 2009 by developing a compactness index to assess changes in the spatial pattern of exurban development over time. Compactness was defined as a measure of how clustered exurban development was in the area surrounding each survey stop at each time period considered. We used Threshold Indicator Taxa Analysis to detect the response of forest and forest-edge species in terms of occurrence and relative abundance along the compactness gradient at two spatial scales (400-m and 1-km radius buffer). Our results showed that most forest birds and some forest-edge species were positively associated with high levels of compactness at the larger spatial scale; the proportion of forest in the surrounding landscape also had a significant effect when forest loss and forest fragmentation were accounted for. In contrast, the spatial configuration of exurban development was an important predictor of occurrence and abundance for only a few species at the smaller spatial scale. The positive response of forest birds to compactness at the larger scale could represent a systematic trajectory of decline and could be highly detrimental to bird diversity if exurban growth continues and creates more compacted development.

## INTRODUCTION

As the world's human population has grown over the last century and residential housing has continued to sprawl even in areas where human population is declining (*Pendall, 2003*; *Seto, Güneralp & Hutyra, 2012*), the rapid increase of housing development has expanded not only at the edge of cities but also beyond the urban fringe to increasingly

more rural areas (e.g., *Davis & Hansen, 2011*; *Hansen et al., 2005*; *Marzluff, 2001*; *McKenzie et al., 2011*; *Suarez-Rubio, Lookingbill & Elmore, 2012*). Housing development beyond the urban fringe (i.e., exurban development) is characterized by low-density, scattered housing units farther away than the suburbs but within commuting distance to an urban center (*Berube et al., 2006*; *Daniels, 1999*; *Lamb, 1983*; *Nelson, 1992*; *Theobald, 2001*). In the conterminous USA, low-density development has been prominent since the 1950s (*Brown et al., 2005*), growing at a rate of about 10% to 15% per year (*Theobald, 2001*). By 2000, 25% of the nation was already considered exurbia (*Brown et al., 2005*) and forecasts have indicated that this pattern of land use will continue into the future (*Brown et al., 2014*; *Kirk, Bolstad & Manson, 2012*).

The attraction of exurban development to areas with high-quality natural and recreational amenities (*Gonzalez-Abraham et al., 2007*; *Hammer et al., 2004*) has raised environmental and ecological concerns (*Gude et al., 2006*; *Hansen et al., 2005*; *Leu, Hanser & Knick, 2008*; *Sampson & DeCoster, 2000*). Exurban development can alter disturbance regimes such as wildfires (*NIFC, 2013*; *Radeloff et al., 2005*) and biogeochemical cycles by changing greenhouse gas fluxes (*Dale et al., 2005*; *Huang, Robinson & Parker, 2014*). By converting natural habitats into exurban development, habitat is lost and fragmented which reduces habitat quality for many native species and increases habitat quality for many early successional and non-native species (*Donnelly & Marzluff, 2006*). In addition to the loss of vegetation cover, changes in structural complexity around houses in exurban areas may have negative impacts on natural communities (*Casey et al., 2009*; *Odell & Knight, 2001*) by degrading habitats and natural resources (*Friesen, Eagles & Mackay, 1995*; *Suarez-Rubio et al., 2013*; *Theobald, Miller & Hobbs, 1997*). As a consequence, exurban development has been linked to reduced survival and reproduction of some wildlife species (*Riley et al., 2003*; *Tewksbury, Hejl & Martin, 1998*) and changes in the behavior and habitat use of other species, for example by interrupting bird migration and movement (*Lepczyk, Mertig & Liu, 2004*; *Miller, Knight & Miller, 1998*).

Forest birds have been found particularly sensitive to new housing (*Pidgeon et al., 2007*) even at densities as low as 0.095 houses/ha (*Friesen, Eagles & Mackay, 1995*; *Merenlender, Reed & Heise, 2009*; *Suarez-Rubio, Renner & Leimgruber, 2011*). Area-sensitive, some cavity-nesting, and bark-foraging birds are relatively more susceptible to the effects of exurban development than granivores, omnivores, and ground foragers (*Fraterrigo & Wiens, 2005*; *Glennon & Kretser, 2013*; *Kluza, Griffin & Degraaf, 2000*; *Merenlender, Reed & Heise, 2009*). Although the mechanisms are not well understood, changes in bird communities have been associated with increased predation (*Engels & Sexton, 1994*; *Lumpkin, Pearson & Turner, 2012*), brood parasitism (*Chace et al., 2003*), free-roaming pets (*Dauphiné & Cooper, 2009*), and activities of landowners (*Lepczyk, Mertig & Liu, 2004*).

The effects of exurban development extend beyond immediate house surroundings. In the Rocky Mountain region of the western USA, an impact zone of up to 180 m from houses has been observed for bird and small-mammal communities (*Odell & Knight, 2001*). Similarly, in the northeastern USA, an ecological effect zone of up to 200 m has been documented for breeding birds (*Glennon & Kretser, 2013*). It is likely that the size of the zone of influence of exurban development is dependent upon the spatial distribution

of houses (*Hansen et al., 2005*). If houses are clustered, the ecological effects of each house overlap, reducing the overall negative impacts. Thus, clustered development is thought to minimize impacts on wildlife habitat relative to highly dispersed low-density housing (*Gagné & Fahrig, 2010*; *Glennon & Kretser, 2013*; *Odell, Theobald & Knight, 2003*; *Theobald, Miller & Hobbs, 1997*). Although the relative importance of habitat quantity over habitat pattern has been shown especially for birds in fragmented systems (*Alberti & Marzluff, 2004*; *Donnelly & Marzluff, 2006*; *Fahrig, 1997*; *Lichstein, Simons & Franzreb, 2002*), little is known about how the spatial pattern of exurban areas changes as this form of development progresses and whether forest birds respond to changes in exurban spatial pattern.

The aim of this study was to assess forest bird responses to changes in the spatial pattern of exurban development and also to examine species responses when forest loss and forest fragmentation were considered. We developed a compactness index to quantify the spatial configuration of exurban development around North American Breeding Bird Survey stops in the Mid-Atlantic region of the USA between 1986 and 2009 and assessed the response of selected bird species (i.e., forest and forest-edge species) along this compactness gradient. In addition, we determined whether species responded differently to exurban patterns at the local (400-m radius buffer) and landscape scale (1-km radius buffer). We hypothesized that exurban development would become more compact over time and thus forest birds would exhibit a decrease in occurrence and relative abundance, whereas forest-edge species would respond positively to compactness of exurban development. To our knowledge, this is the first time that a continuous gradient approach has been used to quantify compactness as exurban development progresses and to identify threshold responses along this gradient.

## MATERIALS AND METHODS

### Study area

Our study area encompassed approximately 4,300 km$^2$ and included nine counties in north-central Virginia (Clarke, Culpeper, Fauquier, Frederick, Madison, Page, Rappahannock, Shenandoah, and Warren) and two in western Maryland (Washington and most of Frederick; Fig. 1), USA. The region has experienced high population growth rates, ranging from 4% (Page County) to 36% (Culpeper County) in the past decade (*US Census Bureau, 2013*). The region has also experienced an increase in exurban settlements over the same time period (*Suarez-Rubio, Lookingbill & Elmore, 2012*), stimulated at least in part by the close proximity of natural amenities (*Suarez-Rubio, Lookingbill & Wainger, 2012*).

### Breeding bird survey

Using the North America Breeding Bird Survey (BBS) (*Peterjohn & Sauer, 1994*; *Sauer, Fallon & Johnson, 2003*), a large-scale annual roadside survey to monitor the status and trend of breeding bird populations in the USA and southern Canada, we selected two groups of species that represent contrasting habitat preferences (forest vs. edge). Forest species—Ovenbird (*Seiurus aurocapilla*), Red-eyed Vireo (*Vireo olivaceus*), American Redstart (*Setophaga ruticilla*), Wood Thrush (*Hylocichla mustelina*), Scarlet

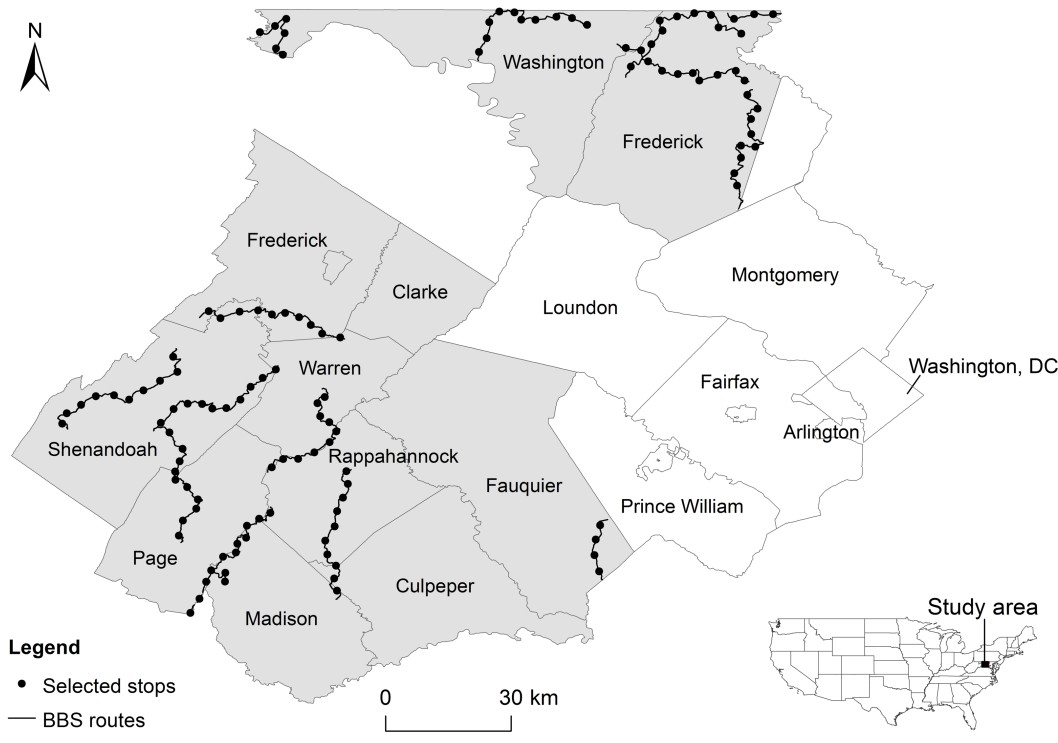

**Figure 1** **Study area (shaded region).** Circles represent 125 North American Breeding Bird Survey (BBS) routes that were uniformly selected from routes.

Tanager (*Piranga olivacea*), and Eastern Wood-Pewee (*Contopus virens*) (*Poole, 2005*)—were defined as birds that use a wide variety of deciduous and mixed deciduous-coniferous forests and that might favor interior forested habitats (*Mikusiñski, Gromadzki & Chylarecki, 2001*). Forest-edge species—Eastern Towhee (*Pipilo erythrophthalmus*), Eastern Phoebe (*Sayornis phoebe*), Gray Catbird (*Dumetella carolinensis*), Northern Cardinal (*Cardinalis cardinalis*), and Indigo Bunting (*Passerina cyanea*) (*Poole, 2005*)—are those species strongly associated with forest edges and open habitats (*Mikusiñski, Gromadzki & Chylarecki, 2001*). These 11 species were also selected because they were detected on at least 5% of surveys during the 1986–2009 interval. In addition, many of the species are reported to have experienced population declines or reduced fecundity due to habitat loss or fragmentation (*Donovan & Flather, 2002*; *HaganIII, 1993*; *Sherry & Holmes, 1997*; *US NABCI Committee, 2009*).

BBS routes involve 39.4 km-long road transects, with 3-minute point count surveys conducted at stops every 0.8 km. From each BBS route located in the study area, we selected every fifth stop along the route to reduce overlap between adjacent areas around survey stops and decrease the likelihood of spatial autocorrelation (Moran's $I = 0.108$, $p = 0.182$). We only considered survey stops that had detailed direction descriptions (i.e., geocoding information and characterization of site-specific features) and fell within the study region (125 survey points in total) (Fig. 1). We focused our analysis on survey stops instead of the entire route because of our interest in local variability of breeding habitats.

To characterize local characteristics of breeding habitats, we established potential zones of influence (*Glennon & Kretser, 2013*) of 400-m and 1-km radius around the selected BBS stops. These areas represented both breeding bird territories (*Bowman, 2003*; *Mazerolle & Hobson, 2004*), which were assumed to be in the immediate surroundings of survey stops, and areas feasibly visited during bird daily movements (*Krementz & Powell, 2000*; *Lang et al., 2002*). Within these areas, we quantified the proportion of forest and exurban development and the spatial pattern of exurban development from 1986 to 2009.

We used a hierarchical Bayesian model to adjust BBS counts (*Suarez-Rubio et al., 2013*) and account for BBS sources of variability such as observer differences (*Sauer, Peterjohn & Link, 1994*), first-year observers' skills (*Erskine, 1978*; *Kendall, Peterjohn & Sauer, 1996*), environmental conditions (*Robbins, Bystrak & Geissler, 1986*), and habitat features (*Sauer, Pendleton & Orsillo, 1995*). We modeled count data as hierarchical over-dispersed Poisson and fit models using Markov Chain Monte Carlo (MCMC) methods in WinBUGS 1.4.3 (*Lunn et al., 2000*). We specified $C_{it}$ as the count for each species on stop $i$ and time $t$ where $i = 1, \ldots, N$; $t = 1, \ldots, T$; and $N$ and $T$ were the number of stops and the number of years species were observed, respectively. $C_{it}$ was assumed to be Poisson distributed with mean $\mu_{it}$

$$C_{it} \sim \text{Pois}(\mu_{it})$$

and the full model was:

$$\log(\mu_{it}) = \beta_{0\text{stop}} + \beta_{1\text{stop}} \times Year_t + \beta_2 \times FirstYear_{it} + Route_{it} + Observer_{it} + Error_{it}$$

where each stop was assumed to have a separate intercept ($\beta_0$) and time trend ($\beta_1$). The model included several sources of variability including unknown route environmental conditions and habitat features ($Route_{it}$), observer effects ($Observer_{it}$), first-year observer effects ($FirstYear_{it}$) and over-dispersion effects ($Error_{it}$). Given that route conditions could also change among years, we also included year into the model. We used two Markov chains for each model and examined model convergence and performance through Gelman–Rubin diagnostics (*Gelman, Carlin & Rubin, 2004*; *Link & Barker, 2010*). Once the model reached convergence, we derived estimates of the count at each stop and in each year which were then used for the threshold analysis.

## Defining exurban development

To characterize the land cover in the areas around survey stops, we classified Landsat 5 TM images (pixel size: 30 m) for 1986, 1993, 2000, and 2009. We performed standard pre-processing procedures (atmospheric and topographic correction) prior to image classification and conducted a supervised classification of areas of exurban development using a training dataset generated from aerial photos. Exurban development was defined as areas with housing densities between 1 unit per 0.4 ha and 1 unit per 16.3 ha (e.g., 6–250 houses per km$^2$) (*Brown et al., 2005*). We identified exurban development using both spectral and structural characteristics following the methods outlined in *Suarez-Rubio, Lookingbill & Elmore (2012)*. We derived spectral characteristics from spectral mixture

analysis (*Adams, Smith & Johnson, 1986*) of the corrected Landsat images to estimate the fractional cover of vegetation, substrate, non-photosynthetic vegetation, and shade within each image. Based on spectral mixture analysis outputs, we built decision trees to classify exurban development for each of the four image dates.

To further analyze pixels belonging to branches of the decision trees that could not discriminate between exurban and urban areas based on spectral characteristics alone, we used morphological spatial pattern analysis (MSPA) (*Soille, 2003*; *Vogt et al., 2007*). The analysis evaluates map geometry by applying mathematical morphological operators to allocate each pixel to one of a mutually exclusive set of classes. We used an 8-neighbor rule as our structural element (i.e., both cardinal directions and diagonal neighbors are considered) and edge width of one. Pixels that fell into the MSPA-Islet (representative of isolated housing units), Bridge, Branch, and Loop classes (representative of associated roads) were considered exurban development. All other MSPA classes were considered urban development. Lastly, all cells originally designated as exurban development in the decision tree were then added back to attain the final exurban development maps. Overall classification accuracy for the final exurban development maps ranged from 93 to 98% (kappa: 0.87–0.96) (*Suarez-Rubio, Lookingbill & Elmore, 2012*).

## Analyzing the spatial pattern of exurban development

To examine the spatial pattern of exurban development, we used the final exurban development maps as foreground and analyzed them using MSPA. Here, we focused specifically on the Islet class which represented scattered, isolated housing units. Using the MSPA classification output, we developed a compactness index to describe how clustered exurban development was in the area surrounding each survey stop at each time period considered. The compactness index was a measure of the proportion of exurban development within any MSPA classes other than the Islet class (i.e., 1−(Exurban Development islets/Exurban Development all classes)) and ranged from 0% (all Islets) to 100% (no Islets). Survey stops lacking exurban development within the potential zone of influence were excluded from the analysis (28 and 20 survey stops for the 400-m and 1-km radius buffers, respectively were excluded). Hence, dispersed exurban development was represented by 0% and maximally clumped exurban development by 100% compactness (see example in Fig. 2).

## Identifying species response to compactness of exurban development

To examine the relationship between compactness of exurban development and bird species at the survey stops, we fitted a non-parametric locally weighted polynomial regression (loess) (*Cleveland & Devlin, 1988*). When the loess regression highlighted nonlinearity in the relationship, then a change-point analysis was used to test for a nonlinear threshold response.

We estimated potential species threshold responses to compactness of exurban development using Threshold Indicator Taxa ANalysis (TITAN) (*Baker & King, 2010*). TITAN allows the identification of change points in both occurrence frequency and relative

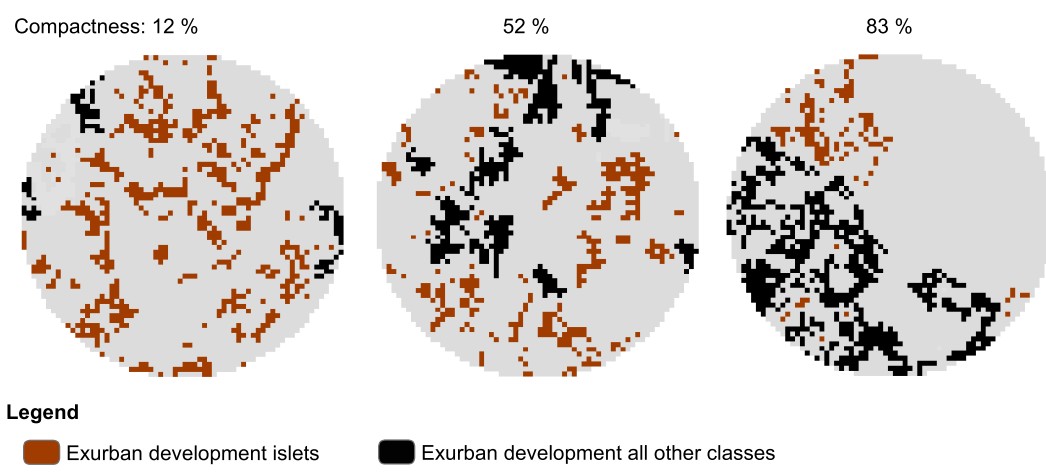

**Legend**

■ Exurban development islets     ■ Exurban development all other classes

**Figure 2  Example of morphological spatial pattern analysis (MSPA) output used to derive level of compactness of exurban development around selected BBS stops.** The illustration shows compactness around 1-km radius buffer of three different BBS stops in 2009 with similar amount of exurban development (20.0 ± 1.3%) among the three landscapes.

abundance of individual species along an environmental gradient. It distinguishes responses of individual species with low occurrence frequencies or highly variable abundances and does not assume a linear response along all or part of an environmental gradient. TITAN uses normalized species scores ($z$) to establish a change-point location that separates the data into two groups and maximizes association of each species with one side of the partition. $Z$ scores measure the association of a species' abundance weighted by their occurrence and are normalized to facilitate cross-species comparison. Thus, TITAN distinguishes if a species responds to an environmental stressor (in this case compactness of exurban development) and whether the response is negative ($z-$) or positive ($z+$).

To measure quality of the response and assess uncertainty around change-point locations, TITAN bootstraps the original dataset and recalculates change points with each simulation. Uncertainty is expressed as quantiles of the change-point distribution. Narrow intervals between upper and lower change-point quantiles (i.e., 5 and 95%) indicate a nonlinear response in species abundance whereas broad quantile intervals are characteristic of species with a linear or more gradual response. Diagnostic indices of the quality of the response are purity and reliability. Purity is the proportion of bootstrap replicates that agree with the direction of the change-point for the observed response. Pure indicators (purity $\geq$ 0.95) are those that consistently assign the same response direction during the resampling procedure. Reliability is the proportion of change-point individual value scores (IndVal) among the bootstrap replicates that consistently have $p$-values below defined probability levels (0.05). Reliable indicators (reliability $\geq$ 0.95) are those with consistently large IndVal.

We ran TITAN (R package: TITAN2) (*Baker & King, 2010*) for the 11 selected bird species and compactness index in R 3.1.1 (*R Development Core Team, 2013*). We used the minimum number of observations on each side of the threshold split that is required by TITAN ($n = 5$) and specified 250 permutations to compute $z$ scores and diagnostic indices as suggested by *Baker & King (2010)*.

### Evaluating species responses to forest loss and forest fragmentation in relation to compactness of exurban development

To evaluate the effects of compactness of exurban development in relation to other factors known to affect birds (i.e., forest loss and forest fragmentation), we used generalized additive models (GAMs) (*Hastie & Tibshirani, 1990*). GAMs were used to better account for potential non-linear trends between the response and predictor variables (e.g., *Guisan, Edwards Jr & Hastie, 2002*; *Zuur et al., 2009*). GAMs require fewer assumptions of data distributions and error structures, assuming only that functions are additive and components can be smoothed by local fitting to subsets of the data.

The models used adjusted counts for each bird species as dependent variables and compactness of exurban development, proportion of exurban development, proportion of forest, number of forest patches greater than 0.45 ha, and forest edge as predictor variables. The latter variables were estimated following *Suarez-Rubio et al. (2013)*. Gaussian errors and an identity link were used, and smoothing parameters were automatically selected based on the effective degrees of freedom and a generalized cross validation criterion in R package mgcv (*Wood, 2001*; *Wood, 2006*). We did a multi-model comparison using a stepwise backwards selection process and calculated the Akaike information criterion ($AIC_i$) and the $\Delta AIC_i$ to rank models and select a best-fitted model (*Zuur et al., 2009*). We used the results to strengthen the inference regarding factors affecting birds in forested environments. Models were evaluated based on graphical diagnostic plots and the explanatory power of a model was assessed by examining the amount of the explained deviance. Predictors of the best-fitted model with high significance levels ($p < 0.01$) were identified as key factors that have strong effects on bird species.

## RESULTS

### Landscape composition and compactness of exurban development around survey stops

Landscape composition around survey stops changed through time during the time period studied, except for the 21% of stops that were inside protected areas (Table 1). The inclusion here of MSPA classes that represented associated roads (i.e., Bridge, Branch, and Loop) in addition to scattered isolated pixels (i.e., Islets) in the definition of exurban development differed from other operational definitions of exurban development used in previous work; as a result, the total amount of development that was classified as exurban was higher for our study than was reported for more restrictive definitions (e.g., *Suarez-Rubio, Lookingbill & Elmore, 2012*). For both the 400-m and 1-km radius buffers, there was a 6% increase in exurban development from 1986 to 2009 (Table 1).

Compactness of exurban development also increased over time (Table 1). For the 400-m radius buffer, compactness increased from 18% in 1986 to 39% in 2009. For the 1-km radius buffer, compactness increased even more, from 11% in 1986 to 44% in 2009. For both extents, the increase was higher between 2000 and 2009 than for any other time period. Compactness was slightly correlated with the amount of exurban development (Pearson's correlation coefficient for 400-m buffer: 0.38, and 1-km buffer: 0.46) and not

**Table 1** Landscape composition and compactness of exurban development (mean ± s.d.) at 400-m and 1-km radius buffer around selected Breeding Bird Survey stops from 1986 to 2009.

| Variables | 1986 | 1993 | 2000 | 2009 |
|---|---|---|---|---|
| **All survey stops** | | | | |
| *400-m radius buffer (n = 97)* | | | | |
| Forest (%) | 34.5 ± 32.3 | 33.6 ± 32.0 | 31.4 ± 31.0 | 24.9 ± 27.2 |
| Exurban development (%) | 11.4 ± 6.5 | 12.1 ± 6.6 | 13.4 ± 6.9 | 17.6 ± 9.4 |
| Compactness (%) | 17.6 ± 26.3 | 18.1 ± 25.8 | 25.1 ± 28.8 | 38.9 ± 34.3 |
| *1-km radius buffer (n = 105)* | | | | |
| Forest (%) | 41.2 ± 30.9 | 40.1 ± 30.5 | 38.5 ± 30.3 | 32.4 ± 28.6 |
| Exurban development (%) | 10.0 ± 4.6 | 10.9 ± 4.8 | 12.1 ± 5.3 | 16.1 ± 7.4 |
| Compactness (%) | 11.2 ± 12.6 | 13.6 ± 13.3 | 23.2 ± 18.0 | 43.9 ± 23.5 |
| **Survey stops in protected area (n = 26)** | | | | |
| *400-m radius buffer* | | | | |
| Forest (%) | 100.0 ± 0.0 | 100.0 ± 0.0 | 99.9 ± 0.4 | 99.9 ± 0.4 |
| Exurban development (%) | 0.0 ± 0.0 | 0.0 ± 0.0 | 0.0 ± 0.0 | 0.1 ± 0.3 |
| *1-km radius buffer* | | | | |
| Forest (%) | 98.7 ± 3.5 | 98.7 ± 3.7 | 98.6 ± 3.8 | 98.1 ± 4.5 |
| Exurban development (%) | 0.3 ± 1.0 | 0.3 ± 1.0 | 0.4 ± 1.1 | 0.7 ± 1.8 |

correlated with forest at either extent (Pearson's correlation coefficient for 400-m buffer: −0.15, 1-km buffer: 0.04).

## Response of bird species to compactness of exurban development

Non-parametric locally weighted polynomial regression (loess) models indicated a non-linear relationship between the compactness index and abundance of selected bird species (Fig. 3). Forest species differed in their threshold response to compactness of exurban development (Fig. 4). For the 400-m radius buffer, only one of the six forest species (i.e., Scarlet Tanager) showed a significant and reliable threshold response to compactness. Although Wood Thrush also responded negatively, the quality of the indicator was less reliable (0.80) (Table 2). In contrast, for the 1-km radius buffer, almost all forest species responded positively and reliably to the compactness of exurban development (Table 2).

Forest-edge species also had significant though less consistent threshold responses to compactness of exurban development at both extents (Fig. 4). For the 400-m radius buffer, Eastern Phoebe and Gray Catbird had a significant positive response to the compactness metric, while Eastern Towhee responded negatively to compactness. For the 1-km radius buffer, Eastern Phoebe, Gray Catbird, and Indigo Bunting responded positively to compactness, with reliability values and change points spanning a wide range of compactness values, similar to the finding for forest birds (e.g., Red-eyed Vireo, Eastern Wood-Pewee; Fig. 4).

In general, reliability information was redundant with purity (i.e., species with ≥0.95 purity were usually also reliable) (Table 2). In some instances, the direction of the response changed with extent of analysis. Wood Thrush responded positively to compactness of exurban development for the 1-km radius buffer. Although the direction of the response

A.

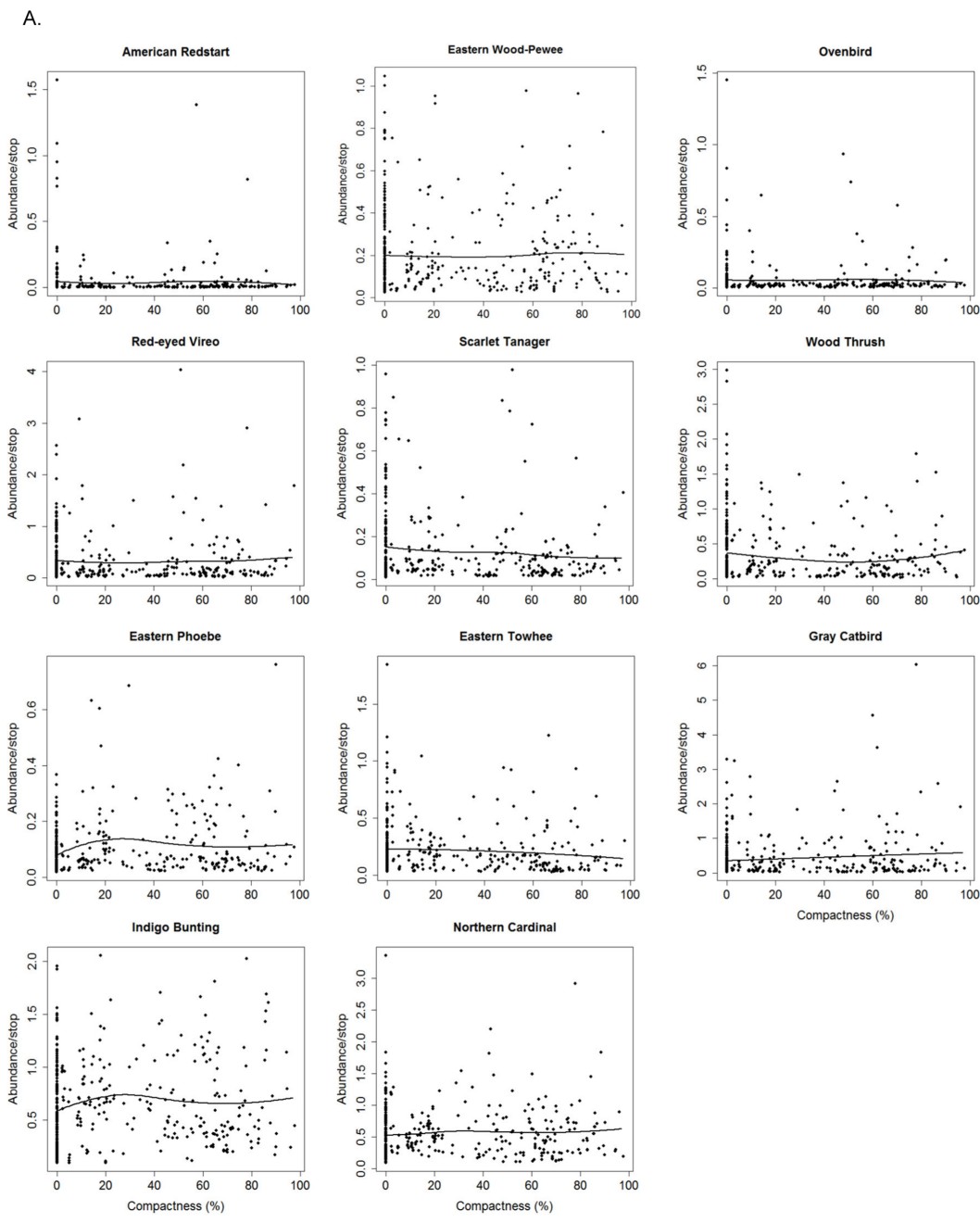

**Figure 3** Relationships between compactness of exurban development and adjusted counts of selected bird species for (A) 400-m and (B) 1-km radius buffer around BBS stops. (continued on next page...)

changed for the 400-m radius buffer, the indicator was not reliable at this extent (reliability = 0.80). For other species (e.g., Scarlet Tanager and Eastern Towhee), wide confidence bands and low $z$ scores at the 400-m extent, highlighted uncertainty when the abundance distributions did not show a clear response. Therefore, where there were differences in the reliability and direction of response at different extents, the 1-km relationships were more reliable.

B.

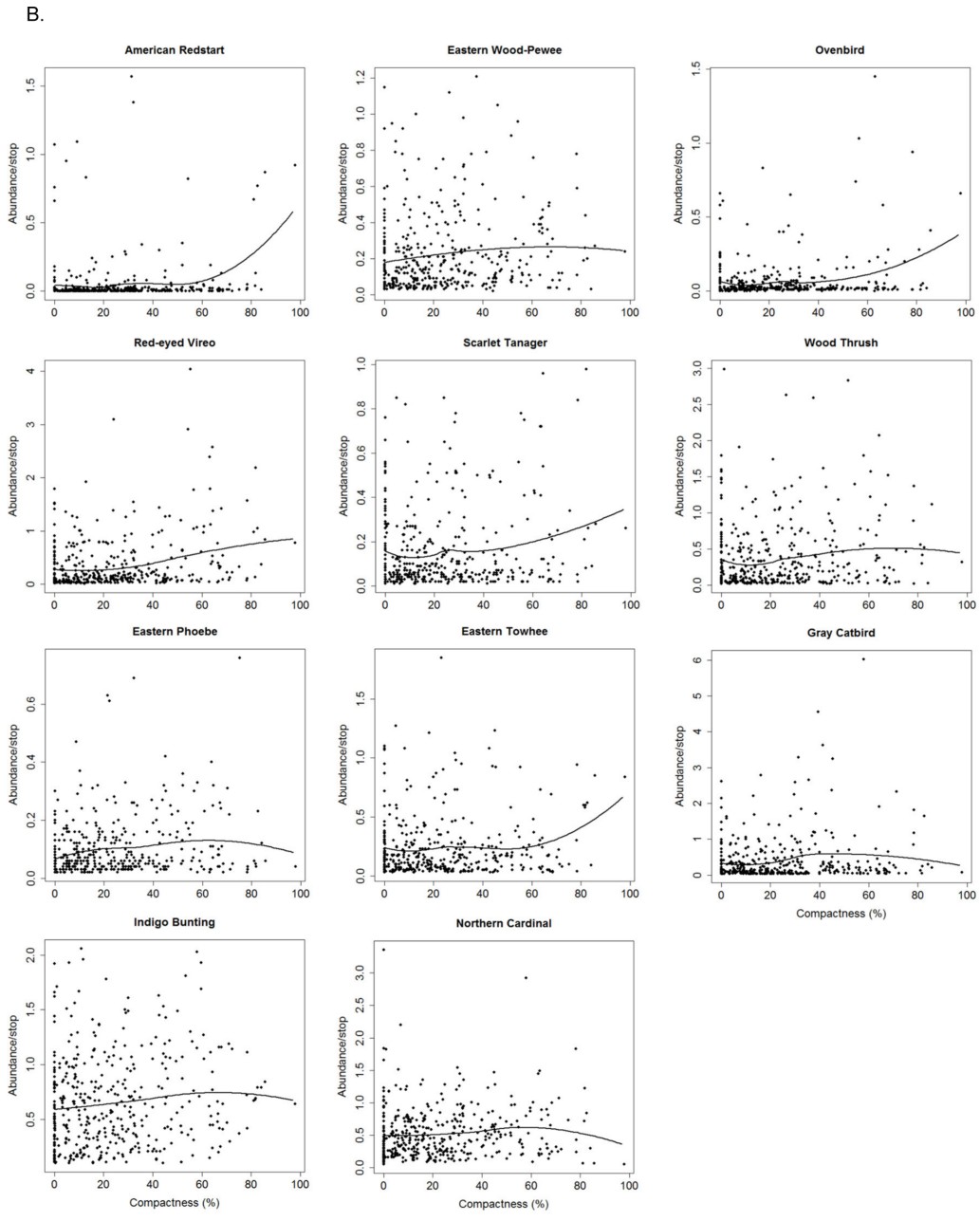

**Figure 3 (...continued)**

Most species (both forest and forest-edge) had relatively broad bootstrapped change-point distributions indicating that there were not sharp threshold responses to compactness of exurban development (Fig. 4). In addition, the width of the bootstrapped change-point distributions varied between the two buffer distances for only a few species. For example, Eastern Phoebe was one of the few species with a sharp response to compactness, but this occurred only at the 400-m radius buffer.

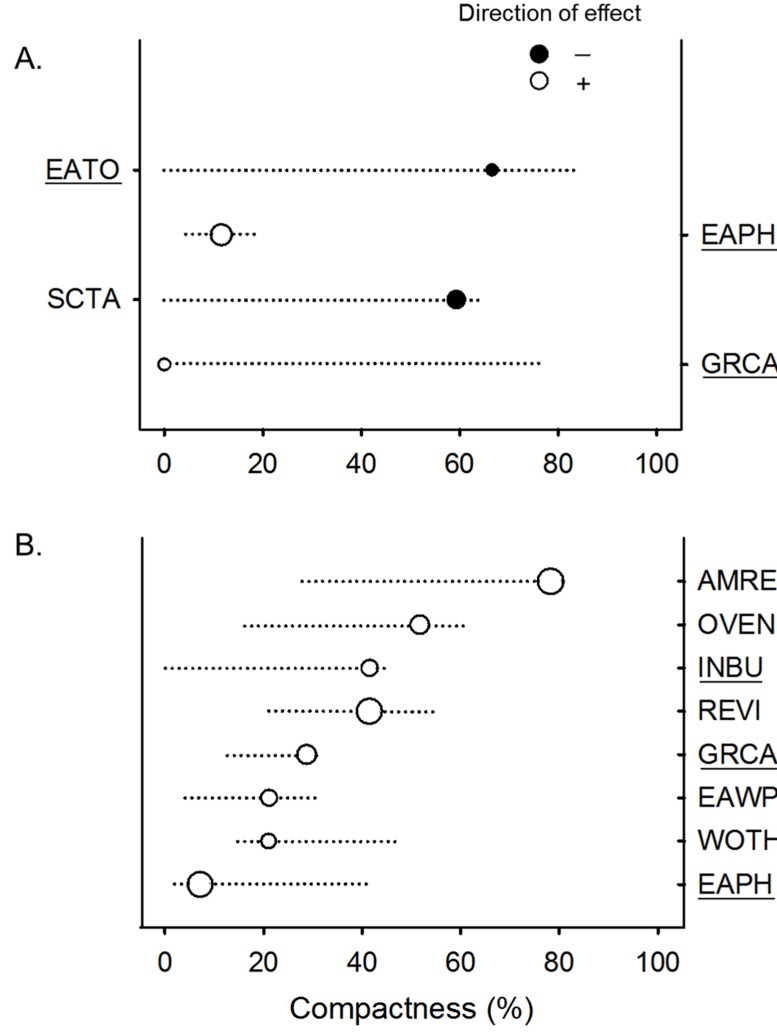

**Figure 4** Change points of significant ($p < 0.05$) and reliable (purity $\geq 0.90$ and reliability $\geq 0.90$) indicator bird species of compactness of exurban development for (A) 400-m and (B) 1-km radius buffer around selected BBS stops. Solid circles represent negative response to compactness (with corresponding species on the left axes) and open circles correspond to a positive response (with corresponding species on the right axes). Circles are sized based on *z* scores and lines represent the 5 and 95% percentiles among bootstrap replicates. Short lines indicate nonlinear response, whereas long lines represent linear or more gradual response. Taxa IDs correspond to American Redstart (AMRE), Eastern Wood-Pewee (EAWP), Ovenbird (OVEN), Red-eyed Vireo (REVI), Scarlet Tanager (SCTA), Wood Thrush (WOTH), Eastern Phoebe (EAPH), Eastern Towhee (EATO), Gray Catbird (GRCA), Indigo Bunting (INBU), and Northern Cardinal (NOCA). Underlined codes denote forest-edge species.

## Response of bird species to forest loss and forest fragmentation in relation to compactness of exurban development

When forest loss and forest fragmentation were included as predictor variables in addition to the exurban development measures (i.e., proportion and compactness), forest had a highly significant effect on all forest species modeled and most forest-edge species at the 1-km radius buffer (Table 3). Number of forest patches had a significant influence on Red-eyed Vireo and Scarlet Tanager, and forest edge did not affect any of the forest species.

**Table 2 Threshold Indicator Taxa ANalysis (TITAN) results for the compactness index at the 400-m and 1-km radius buffer.** Significant ($p < 0.05$) and reliable (purity $\geq 0.90$ and reliability $\geq 0.90$) species are shown in bold.

| Species | Direction of effect | z | Change point | | | Purity | Reliability | p |
|---|---|---|---|---|---|---|---|---|
| | | | Obs. | 5% | 95% | | | |
| **400-m radius buffer** | | | | | | | | |
| *Forest birds* | | | | | | | | |
| AMRE | − | 0.94 | 0.00 | 0.00 | 84.92 | 0.54 | 0.31 | 0.180 |
| EAWP | − | 1.28 | 89.19 | 0.00 | 89.58 | 0.54 | 0.47 | 0.116 |
| OVEN | − | 1.84 | 0.00 | 0.00 | 87.40 | 0.59 | 0.38 | 0.052 |
| REVI | − | 1.52 | 0.00 | 0.00 | 86.16 | 0.56 | 0.40 | 0.072 |
| **SCTA** | − | 4.85 | 59.33 | 0.00 | 64.09 | 1.00 | 0.99 | 0.004 |
| WOTH | − | 3.00 | 18.81 | 0.00 | 77.75 | 0.81 | 0.80 | 0.012 |
| *Forest-edge species* | | | | | | | | |
| **EAPH** | + | 5.81 | 11.57 | 4.40 | 19.30 | 0.98 | 0.98 | 0.004 |
| **EATO** | − | 3.06 | 66.60 | 0.00 | 82.98 | 0.93 | 0.91 | 0.004 |
| **GRCA** | + | 3.26 | 0.00 | 0.00 | 78.92 | 0.96 | 0.94 | 0.008 |
| INBU | + | 3.41 | 9.05 | 0.00 | 85.84 | 0.90 | 0.89 | 0.008 |
| NOCA | + | 1.95 | 74.91 | 0.00 | 89.19 | 0.80 | 0.71 | 0.056 |
| **1-km radius buffer** | | | | | | | | |
| *Forest birds* | | | | | | | | |
| **AMRE** | + | 7.03 | 78.26 | 27.58 | 80.66 | 1.00 | 1.00 | 0.004 |
| **EAWP** | + | 4.45 | 21.11 | 4.00 | 31.27 | 0.99 | 0.98 | 0.004 |
| **OVEN** | + | 5.16 | 51.70 | 16.07 | 61.89 | 0.99 | 0.99 | 0.004 |
| **REVI** | + | 6.99 | 41.47 | 20.98 | 55.16 | 1.00 | 1.00 | 0.004 |
| SCTA | + | 3.92 | 53.86 | 0.00 | 60.16 | 0.89 | 0.89 | 0.008 |
| **WOTH** | + | 4.06 | 20.98 | 14.98 | 47.12 | 0.97 | 0.96 | 0.004 |
| *Forest-edge species* | | | | | | | | |
| **EAPH** | + | 6.86 | 7.15 | 1.85 | 41.76 | 1.00 | 1.00 | 0.004 |
| EATO | + | 2.73 | 78.26 | 0.00 | 81.38 | 0.86 | 0.84 | 0.016 |
| **GRCA** | + | 5.25 | 28.74 | 12.46 | 31.33 | 1.00 | 0.99 | 0.004 |
| **INBU** | + | 4.48 | 41.54 | 0.00 | 45.00 | 0.99 | 0.98 | 0.004 |
| NOCA | + | 4.13 | 28.54 | 0.00 | 81.74 | 0.82 | 0.82 | 0.004 |

The effect of exurban development varied among forest species. Only Red-eyed Vireo was significantly influenced by both proportion of exurban development and compactness of exurban development. Eastern Wood-Pewee and Wood Thrush were influenced by compactness of exurban development, whereas Scarlet Tanager was only influenced by proportion of exurban development.

None of the forest-edge species were influenced by compactness of exurban development at the 1-km radius buffer, although Eastern Phoebe, Eastern Towhee, Indigo Bunting, and Northern Cardinal were affected by its proportion. Regarding forest fragmentation, Indigo Bunting and Northern Cardinal were influenced by number of forest patches, whereas Eastern Phoebe, Eastern Towhee, and Gray Catbird were affected by forest edge. Models

**Table 3 Summary of generalized additive models (GAM) for forest and forest-edge bird species at the 1-km radius buffer.** Only species in which the model was a good fit were included. Smoother is represented by *s* and year was included as a factor in the model therefore a smooth term did not apply. $\Delta AIC_i$ was used to rank models and only full and best-fitted model are shown. Significant values ($p < 0.01$) are shown in bold.

| | | | Forest | Exurban development | Compactness | Forest patches > 0.45 ha | Forest edge | Year | Deviance explained (%) | GCV | $\Delta AIC_i$ |
|---|---|---|---|---|---|---|---|---|---|---|---|
| *Forest birds* | | | | | | | | | | | |
| EAWP | Full | *s* | 6 | 5 | 2 | 1 | 1 | 3 | 30.3 | 0.654 | 2.719 |
| | | *p* | **<0.001** | 0.049 | **0.001** | 0.745 | 0.356 | 0.016 | | | |
| | Best-fitted | *s* | 7 | 5 | 2 | | | 3 | 30.2 | 0.649 | 0 |
| | | *p* | **<0.001** | 0.067 | **<0.001** | | | 0.018 | | | |
| REVI | Full | *s* | 1 | 1 | 1 | 7 | 2 | 3 | 66.5 | 0.554 | 0.120 |
| | | *p* | **<0.001** | **0.004** | **<0.001** | **0.007** | 0.320 | **0.006** | | | |
| | Best-fitted | *s* | 1 | 1 | 1 | 7 | | 3 | 65.2 | 0.554 | 0 |
| | | *p* | **<0.001** | **<0.001** | **<0.001** | **0.008** | | **0.012** | | | |
| SCTA | Full | *s* | 4 | 6 | 1 | 7 | 7 | 3 | 64.1 | 0.453 | 1.297 |
| | | *p* | **<0.001** | **0.002** | 0.464 | **0.002** | 0.091 | 0.914 | | | |
| | Best-fitted | *s* | 4 | 5 | | 7 | 7 | 3 | 64.1 | 0.451 | 0 |
| | | *p* | **<0.001** | **0.003** | | **0.003** | 0.081 | 0.810 | | | |
| WOTH | Full | *s* | 1 | 5 | 2 | 7 | 6 | 3 | 42.0 | 0.999 | 2.955 |
| | | *p* | **<0.001** | 0.091 | **0.006** | 0.013 | 0.094 | 0.585 | | | |
| | Best-fitted | *s* | 4 | 5 | 3 | 7 | | | 40.8 | 0.990 | 0 |
| | | *p* | **<0.001** | 0.039 | **0.005** | 0.012 | | | | | |
| *Forest-edge species* | | | | | | | | | | | |
| EAPH | Full & best-fitted | *s* | 2 | 1 | 3 | 4 | 8 | 3 | 31.8 | 0.506 | 0 |
| | | *p* | **<0.001** | **0.003** | 0.022 | 0.120 | **<0.001** | **0.003** | | | |
| EATO | Full & best-fitted | *s* | 5 | 2 | 1 | 2 | 9 | 3 | 27.9 | 0.679 | 0 |
| | | *p* | **<0.001** | **0.001** | 0.199 | 0.259 | **0.001** | 0.875 | | | |
| GRCA | Full | *s* | 2 | 2 | 5 | 1 | 8 | 3 | 16.7 | 1.520 | 0.435 |
| | | *p* | 0.018 | 0.096 | 0.047 | 0.131 | 0.026 | 0.805 | | | |
| | Best-fitted | *s* | 3 | 3 | 5 | | 8 | 3 | 16.6 | 1.518 | 0 |
| | | *p* | 0.040 | 0.102 | 0.040 | | **0.007** | 0.715 | | | |
| INBU | Full | *s* | 8 | 7 | 1 | 5 | 1 | 3 | 29.5 | 0.415 | 3.484 |
| | | *p* | **<0.001** | **0.001** | 0.233 | **0.006** | 0.234 | 0.634 | | | |
| | Best-fitted | *s* | 7 | 7 | | **5** | | | 28.2 | 0.411 | 0 |
| | | *p* | **<0.001** | **<0.001** | | **0.006** | | | | | |
| NOCA | Full | *s* | 1 | 5 | 1 | 4 | 1 | 3 | 11.6 | 0.462 | 4.823 |
| | | *p* | 0.306 | 0.264 | 0.151 | 0.020 | 0.166 | 0.584 | | | |
| | Best-fitted | *s* | | 5 | | 4 | | | 10.0 | 0.079 | 0 |
| | | *p* | | **0.006** | | **0.009** | | | | | |

at the 400-m buffer and for American Redstart and Ovenbird at the 1-km buffer did not converge.

## DISCUSSION

Our results suggest that both forest birds and some forest-edge species responded to spatial patterns of exurban development at the landscape extent (1-km radius buffer) (Fig. 4B). Contrary to our prediction, forest birds exhibited a positive response to compactness of exurban development with change points between 21% and 78% (Table 2). These results indicate that frequency and abundance of forest birds increase as compactness increases. There are a few potential explanations for this pattern. First, although compactness of exurban development increased over time, these bird species were also increasing in abundance generally in the region (*Suarez-Rubio et al., 2013*) partly due to forest regrowth (*Bowen et al., 2007*) and protected areas adjacent to the study area. Second, forest disturbance associated with exurban development may benefit forest birds, especially forest birds such as American Redstart and Red-eyed Vireo that seem to occur more frequently in early and mid-successional forests and even start to decline as forests mature (*Holmes & Sherry, 2001*; *Hunt, 1998*). Lastly, even though forest decreased around survey stops, forest cover was nonetheless above the minimum amount of habitat necessary for the persistence of forest birds (>30%; *Andrén, 1994*; *Betts, Forbes & Diamond, 2007*; *Radford, Bennett & Cheers, 2005*; *Suarez-Rubio et al., 2013*; *Zuckerberg & Porter, 2010*).

When the effects of compactness of exurban development were assessed in relation to forest loss and forest fragmentation, proportion of forest had a highly significant effect compared to compactness in most cases (Table 3). This indicates that for forest birds, proportion of forest at the landscape extent may be more important than exurban development. However, proportion of exurban development and compactness also had a significant effect, which suggests that if proportion of exurban development or compactness continues this would inevitably lead to the loss of forest species.

Surprisingly, Indigo Bunting and Gray Catbird (i.e., forest-edge species) also responded positively to compactness of exurban development at the landscape extent with change points similar to those exhibited by forest birds (Table 2 and Fig. 4B). Although Indigo Bunting is known for its strong preference for edges, and surely human habitat modification (e.g., clearing of woods) increases suitable habitat for buntings (*Payne, 2006*), bunting numbers have declined in eastern North America since the last quarter of the twentieth century (*Sauer et al., 2014*). These declines have been associated with increasing levels of brood parasitism and predation that occur in fragmented habitats (*Donovan & Flather, 2002*; *Robinson et al., 1995*) but also with forest regrowth, which has reduced shrubby habitats that they tend to use (*DeGraaf & Yamasaki, 2003*). It is important to note that when forest loss and forest fragmentation were also considered, the effect of compactness was not significant and proportion of forest and exurban development had a greater influence. This suggests that buntings may be more sensitive to habitat quantity than the spatial pattern of exurban development.

Gray Catbird is frequently associated with suburbia and also prefers early successional habitats, and shrubs around houses have probably increased the availability of breeding

habitat for this species (*Smith et al., 2011*). Although compact exurban development may minimize the disturbance associated with domestic predators introduced in exurban areas that usually prey directly on nests (*Balogh, Ryder & Marra, 2011*; *Lepczyk, Mertig & Liu, 2003*; *Lumpkin, Pearson & Turner, 2012*), the effects of compactness diminished when forest loss and fragmentation were also taken into account at the landscape extent.

At the local extent (i.e., 400-m radius buffer), Scarlet Tanager responded negatively, whereas Gray Catbird responded positively to compactness of exurban development, with both exhibiting gradual responses (Fig. 4A). Scarlet Tanager is an interior forest species that is very sensitive to forest fragmentation (*Rosenberg, Lowe & Dhondt, 1999*). In a previous study, this species was found to have a negative response to the amount of exurban development at very low levels (*Suarez-Rubio et al., 2013*). Thus, Scarlet Tanager appears to be negatively affected by exurban development regardless of its spatial configuration, which was also the case for the landscape extent. The positive response of Gray Catbird to compactness of exurban development perhaps indicates that predation pressure by introduced domestic predators in exurban areas (*Lepczyk, Mertig & Liu, 2003*; *Lumpkin, Pearson & Turner, 2012*) affects catbirds at the local extent. Exurban areas have large numbers of non-native plant species (*Gavier-Pizarro et al., 2010*; *Lenth, Knight & Gilgert, 2006*; *Maestas, Knight & Gilgert, 2003*), and there is some evidence that nests in exotic shrubs are twice as likely to be depredated and suffer higher rates of nest failure than nests in native shrubs (*Borgmann & Rodewald, 2004*), although this is not always the case (*Meyer, Schmidt & Robertson, 2015*).

Interestingly, most forest birds did not exhibit threshold responses to compactness of exurban development at the local extent. This difference in response at the local and landscape extent suggests that the effects of compactness of exurban development are scale dependent. *Smith, Fahrig & Francis (2011)* demonstrated that effects of fragmentation change with the extent of analysis because ecological processes (e.g., predation) act at different spatial scales. Thus, the effects of compactness of exurban development might be associated with the size of the disturbance zone. Other studies have found an ecological effect zone of up to 200 m from exurban homes in which avian densities were altered (*Glennon & Kretser, 2013*; *Odell & Knight, 2001*).

Our results reveal that the responses of forest birds varied, but extended well beyond a 200-m radius. When considering a 400-m zone of influence, most forest birds did not respond significantly to the spatial pattern of exurban development. However, the spatial compactness of development was associated with a positive response at the 1-km zone for nearly all forest bird species. Previous studies have shown that forest birds are very sensitive to the proportion of exurban development (e.g., *Pidgeon et al., 2007*; *Suarez-Rubio et al., 2013*). Our results show that forest birds are also sensitive to its spatial configuration at large extents. In general, if exurban development occurs in the landscape, it affects the entire 400-m radius buffer regardless of its arrangement, but by aggregating exurban development within the 1-km radius buffer, safe zones were retained that could support forest birds and the effects of compactness of exurban development were reduced.

By assessing the spatial pattern of exurban development for the multiple images, we were able to capture the dynamics of landscape change over time (Table 1) as was also

done previously for the conterminous United States (e.g., *Mockrin et al., 2012*; *Pidgeon et al., 2014*). As exurban areas grew, scattered, isolated exurban development became more contiguous and clumped. Thus, our results demonstrate the effects of the spatial pattern of exurban development within the larger context of forest habitat loss. At the level of individual survey stops, the positive but weak correlation between exurban development and compactness indicates that there is variance in spatial configuration that is independent from the overall amount of exurban development.

Although the total amount of exurban development around survey stops increased compared to previous operational definitions (*Suarez-Rubio et al., 2013*), forest loss and forest fragmentation did not vary when definitions were compared (Appendix S1). Thus, by including both isolated and scattered housing units and associated roads into our definition, we were able to reflect the substantial expansion of exurban development that has occurred in the region (e.g., *Suarez-Rubio, Lookingbill & Elmore, 2012*). In addition, by considering the effects of the spatial pattern of exurban development together with forest loss and forest fragmentation, we identified the importance of compactness in light of other factors that are known to affect forest birds.

Nonetheless, some caveats arise. The use of bird counts along BBS routes may not fully reflect occurrence and abundance of more sensitive species such as Kentucky Warbler. Although counts along roadsides have been shown to be representative of changes occurring over much broader areas (*Keller & Scallan, 1999*), our findings cannot be generalized beyond the range of housing density included in this study (e.g., to wilder or more urbanized areas). In addition, the compactness index was developed to assess the clumpiness of exurban housing and assumed presence of housing units; thus it is not suitable for comparison to areas without development.

A critical unknown of exurban growth is the possible cumulative impacts on wildlife. Evaluating potential cumulative impacts requires an enhanced understanding of both the density and patterns of residential development and of the distinct effects of these two components of landscape change (*Pidgeon et al., 2014*; *Theobald, Miller & Hobbs, 1997*). We have taken a first step by identifying the extent at which forest and forest-edge species respond to the spatial patterning of exurban development and highlight that the positive response of forest birds to compactness at the larger extent should be viewed with caution in the larger context of a systematic trajectory of bird diversity decline (*Pidgeon et al., 2014*). If exurban growth continues to increase, as trends suggest, this will lead towards more contagious development. Thus, management efforts should try to concentrate development away from ecological sensitive areas, create or maintain safe zones, and minimize forest loss or fragmentation (i.e., increase compactness) to support forest birds.

## ACKNOWLEDGEMENTS

The authors thank thousands of volunteers who have collected Breeding Bird Survey Data and D Ziolkowski and K Pardieck (USFWS) for providing the bird data, the topographic maps, and the description of the BBS stops. S Wilson and R Hildebrand

provided helpful analytic advice. We thank C Elphick, C Rittenhouse, and anonymous reviewers for comments that greatly improved the manuscript.

### Funding
The authors received no funding for this work.

### Competing Interests
The authors declare there are no competing interests.

### Author Contributions
- Marcela Suarez-Rubio conceived and designed the experiments, performed the experiments, analyzed the data, contributed reagents/materials/analysis tools, wrote the paper, prepared figures and/or tables, reviewed drafts of the paper.
- Todd R. Lookingbill conceived and designed the experiments, contributed reagents/materials/analysis tools, wrote the paper, reviewed drafts of the paper.

### Data Availability
    The raw data was submitted as Data S1.

### Supplemental Information
Supplemental information for this article can be found online at http://dx.doi.org/10.7717/peerj.2039#supplemental-information.

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
