# Peer review of "Forest birds respond to the spatial pattern of exurban development in the Mid-Atlantic region, USA"

_PeerJ, doi:10.7717/peerj.2039_

## Round 0.1 · original submission · Major Revisions

· Academic Editor

Major Revisions

This is an interesting paper and both reviewers saw value in the work. Reviewer 2 outlines some additions to the current analysis, that I think are necessary in terms of knowing how to interpret the results. Below I outline some comments based on my own read of the paper.

Line 123: Although the method used here might reduce the risk of autocorrelation, there could still be spatial structure in the data. Is there justification for subsampling at this scale (e.g., analyses done with these routes, or other work using BBS data)? If not, either testing for autocorrelation or adding it into the models would be helpful.

Line 163: I would specify the pixel size. Many readers will know, but it might be helpful for those unfamiliar with the imagery.

Line 215: Please specify the R package. (I assume this analysis is not in base R and package titan doesn’t look right …)

Line 238: Why give only examples in Fig 3? Since space is not limiting at PeerJ, I see no reason not to provide the results for all species.

Also, perhaps I missed it, but did you directly compare these models to linear alternatives. What evidence is there that the nonlinear models provide better fit to the data?

Line 242: I concur with reviewer 2 in my uncertainty about what is meant here. Is it that you mean only 1 of 6 forest indicator species were both significant and reliable (i.e., SCTA, WOTH, etc. are forest indicators)? If so, maybe you could just drop the word "indicator" throughout the paper, since it creates confusion given its variable and often ambiguous use in the literature. If the intended meaning is different, please clarify.

Line 252: By which forest species? There was a range of values for these measures across the forest species.

Line 258-9: Maybe I am misreading the table, but it looks like 3 of 10 cases involve opposite results at the two scales. Moreover, could the situation for some of the other cases be unclear because of uncertainty in the estimate of z (I don’t see that uncertainty reported); i.e., can you be sure that those cases do not also have a difference that went undetected? For the 3 cases with differences, SCTA and EATO are more reliable at the 400-m scale, while WOTH is more reliable at the 1-km scale. My interpretation does not seem to match what is written in this paragraph. Am I just not understanding Table 2?

Line 274: Were these areas increasingly intact (i.e., through regrowth). If not, I’m not sure how they could account for an increase in forest bird abundance. Simply remaining intact would only account for a lack of decrease.

Line 282: The logic behind this argument is unclear to me, because the forest interior species described in this study are not the same species that come into feeders, plantings, etc. and that account for the increase in richness with low level development. I agree with reviewer 2’s suggestion that other variables may be confounding the analysis and that forest loss and fragmentation should be considered as covariates.

Also, the data for the species in Fig 3 suggests that any biological change over the compactness scale has a pretty low magnitude, in relation to the variation in the data. This suggests that the biological effects are very small. Given the other things that potentially affect bird densities, I wonder how biologically important such effects are? Evaluating their importance in light of other things known to affect forest birds would thus seem especially important.

Line 293: Indigo bunting declines in the eastern US are also almost certainly attributed to forest regrowth, which has reduced the shrubby habitats that they tend to use. Note that there is a big difference between catbird and indigo buntings. The former are frequently associated with suburbia, whereas the latter are not.

Fig. 4. Putting species names on both vertical axes seems potentially confusing given that you have two panels side by side. Since you clearly indicate direction of effect with the symbol, I would put them all on the left axis. Alternatively, you could stack the panels on top of each other.

Reviewer 1 ·

Basic reporting

My comments are focused on the use of clear and unambiguous text in the manuscript. A quantitative metric such as compactness and how it changes over time and at different scales does not lend itself to an intuitive interpretation of how, what or why birds might be responding in certain ways within the landscape. Because the language used in the methods section was quite technical, it was challenging to interpret if and how each test was designed to address a potential biological change or a mechanism driving avian response. Although the authors may wish to retain this technical language, statements relating methodological approaches to the biology of how or why birds might be expected to respond to changes in compactness or overall level of development would be helpful. This would also help the reader develop a deeper understanding of the potential mechanisms driving the results, which were contrary to the authors’ predicted response.

In the discussion session, it would be helpful if you could place your results within the larger landscape context. What changes in land use cover are happening at the regional or flyway scale that could be contributing to your results? Are there any limitations or biases associated with BBS survey data that could influence your results that the reader should be aware of?

Figure 4 – the caption for this figure is quite long and includes information that should be limited to the methods section. Or, possibly there was a formatting error? There are two paragraphs in my version, one that is quite brief, the other quite long.

Experimental design

As relates to the knowledge gap being filled and how this study contributes to filling that gap, I was surprised that the authors failed to mention the large body of work that Dr. John Marzluff and his students and collaborators have produced on the subject of avian responses to urban development, including exurban development. By not referencing this body of work, I’m concerned that the authors might have missed an opportunity to place their research in the context of the current state of knowledge on the subject.

Validity of the findings

No Comments

Comments for the author

This study contributes to a growing body of work documenting avian response to exurban development, identifying threshold responses for species at difference scales of interest. For species that were predicted to decline in response to increased compactness but instead had a positive association, these results suggest that further research on the topic should focus on the specific elements of the landscape that birds are responding to, e.g., increased food availability, human development pressure in surrounding areas, etc., and that it is important to consider multiple scales when assessing avian response to landscape change.
If the authors are interested in promoting the application of these results in land use planning it would be helpful if they could provide a more straightforward interpretation of how this research might guide the development of more bird-friendly suburbs and exurban areas.

·

Basic reporting

The article is written well with respect to structure and flow. Some terms require additional or revised explanations to improve clarity, e.g., Compactness index, indicator taxa, indicator response taxa, and indicator response.

Line 180-183: Compactness index. I re-read the description several times, and dwelt on figure 2. I’m having a hard time lining up the text with figure 2. The best I can interpret from the figure, the compactness index is:
MSPA all other classes / (MSPA islet + MSPA all other classes)
Which would make low values for compactness when MSPA all other classes is low proportion of the landscape. This is not what the text says. Please clarify.

Line 242: Term “indicator taxa” – after reading this statement and going back to the description in lines 193 – 202, I’m still lost as to what “indicator taxa” and “indicator response taxa” and “indicator response” means. Coming from the conservation arena, I read indicator taxa and think indicator species, meaning a species whose presence or abundance is representative of certain environmental conditions. Please re-write the explanation of these terms with the goal of moving them from jargon to technical terms.

Experimental design

The experimental design is sufficient for the question at hand. My comments primarily seek clarification of the methods.

Line 143 and 146: I see Cit as count for species at stop i at time t, but I don’t see where Cit is in the model statement on line 146.

Line 156: Was classification accuracy assessed? How? And what are the results?

Line 167: Which imagery was classified with MSPA? Was it the (pre-processed) Landsat 5 TM images? The classified Landsat 5 Tm images from the previous paragraph?

Line 167: I’m familiar with MSPA and have used it in analyses myself. This is a clever application. I see why the edge width of 1 cell (30m pixel) was used – to grab only the isolated cells – and why only certain MSPA classes were considered exurban. Nicely done. A question though, how sensitive is compactness to a change in the edge definition? In the discussion. Lines 313 to 321, the literature suggests edge effects extend much more than 30 m. Where any other edge definitions (e.g., 60 or 90 m) explored?

Line 202: negative (z-) and positive (z+) indicator response taxa…it’s not immediately clear that z is the “effect” of interest and the sign indicates the direction of the effect, e.g., (–) means the species has a negative response to increasing compactness. In Table 2 and Figure 4, is it appropriate to say “Direction of effect” and use the sign, omitting z?

Validity of the findings

My primary concern is in the validity of the findings in light of alternative explanations for the patterns observed here.

The authors examined forest and forest-edge bird response to compactness of housing development. The definition of exurban is good, as are the methods to identify it after addressing clarifying comments.

However, I’m curious as to how factors known to affect forest birds – forest loss, forest fragmentation, and forest degradation – are treated in the analysis. It seems to me that exurban development may be confounded with forest loss (the houses were built on something, likely forest?) and with forest fragmentation (decrease in patch size, increase in edge without change in forest area).

Without accounting for loss or fragmentation in the analysis, an alternative explanation for the results is that minimizing forest loss (increasing compactness) or minimizing forest fragmentation (increasing compactness) are beneficial for forest birds.

Please address forest loss and fragmentation in the analysis. This might be done by incorporating the information from Table 1 in the analysis of species responses to compactness. The introduction should also provide a brief literature review on forest loss vs fragmentation, and the discussion revised to place the results into the broader context of exurban development in forested environments.

Comments for the author

Overall, this is well-written, well thought-out study addressing a highly relevant question: does compact exurban development reduce negative impacts on forest birds?

The statistical approaches to address this question are top notch, though not complete. Please address how forest loss, forest fragmentation, and forest degradation may be confounding factors in the compactness analysis. This may be accomplished by incorporating the information from Table 1 in the analysis of species responses to compactness.

---

## Round 0.2 · Major Revisions

· Academic Editor

Major Revisions

Thank you for your careful revision and response letter. One of the original reviewers and I have re-read the paper and find it much improved. We both have additional comments about the analyses, however, and I would like to see them addressed before I make a final decision.

First, I remain confused about how the different analyses relate to one another and concerned that what could be a fairly simple analysis has become overly complicated with multiple techniques overlapping one another. Maybe it is because I am not very familiar with all of the methods you are using (i.e., TITAN), but I’m having trouble understanding the relationship between the hierarchical Bayesian model described in the paragraph starting on line 160 and the subsequent analyses. Is it that you are only using this model to obtain adjusted values, then drawing (adjusted) values from the posterior distributions for use in subsequent analyses? And, if that’s the case, am I correct that the subsequent analyses are not Bayesian? If I am right about this, then I’m unclear why one would not simply add the potentially explanatory variables (forest loss, compactness, etc.) to the Bayesian model and do everything in a single analysis. Please do not hesitate to explain if I am just misunderstanding the approach you have taken. But, the apparent approach of using one analysis to adjust the dependent variables, then another to look for thresholds, then yet another to look at the relative importance of different variables, seems unnecessarily complex.

Second, the reviewer brings up a related point in their comments on multiple model comparisons. Please address this concern either by using a model comparison approach, as suggested, or by providing a detailed rationale for not doing so (e.g., in a fully Bayesian model it may not be necessary, depending on how the model is set up).

Please also address the additional comments below and the other comments from the reviewer. I have also attached a pdf with some minor wording edits.

Line 24: I don’t find this definition of compactness to be as helpful as it might be, because it just begs the question of what is “clumpiness”. Please define in terms that explain the spatial patterning. Maybe something like this “… as a measure of how clustered exurban development was in the area surrounding …”

Line 29 (and 33): “Landscape” vs “local” are relative terms that will likely vary in their meaning among species with different home range sizes. I would suggest “compactness at the larger spatial scale …” instead. In the body of the paper itself, the terms are formally defined, so the change is less important there than in the abstract.

Line 30: I’d suggest “although the proportion of forest in the surrounding landscape had” instead, though please check that this is what was meant.

Line 141: Although the BBS data are familiar to many, I’d suggest adding a sentence to explain the basic structure – some readers might otherwise not know what comments like “every fifth stop” refer to. Something along the lines of “BBS routes involve 24.5 mile-long road transects, with 3-minute point count surveys conducted at stops every 0.5 miles.”

Line 162: It’s not clear to me how this statement corresponds to the variables given in the full model. In particular, where are the habitat features and environmental conditions in that model (in other words, is it really a full model, or is it a base model to which potentially explanatory variables were added)? And what variables were chosen, and why? Related to this, is “Noise” (in the equation) a measure of environmental noise … or is it, as I first assumed, a synonym for error. (Note that my questions about this part of the analysis may be irrelevant depending on your response to my general question about the analysis, earlier in these comments.)

Line 316. I’m not sure I would say that the change points are similar to forest species. If I’m reading the figure right, it looks as though both groups have change points that just span a wide range of compactness values – i.e., there is no real pattern in either group. I think that is perhaps what you are suggesting, but the wording implies something else.

Line 328. I think it might be clearer to say “indicating that there were not sharp threshold responses to compactness of …”.

In the legend to Fig 4 please give the actual acronym definitions rather than just referring to the AOU. Also in this figure it is not clear to me what circle size indicates.

·

Basic reporting

Overall the manuscript is greatly improved. I have 2 line comments:

Line 89-91: “The aim of this study was to assess forest bird response to changes in the spatial pattern of exurban development, and to examine species response when also forest loss, and forest fragmentation.”
Line 189: Add “compactness”, e.g., “The compactness index…”
Line 347: “the effect of compactness was reduced…” how is the reduction in effect size determined?

Experimental design

No new comments.

Validity of the findings

The 1, 2 approach of first looking at compactness index and then looking at compactness with forest variables is interesting. Contemporary approaches to model fitting would suggest a form of multi-model comparison rather than fitting a full model only. Also, there are formal ways to assess the relative effects of different factors fit in the same model beside p-value alone. An example given the full model approach would be to standardize variables so they have the same basis for comparison, and then comparing the size of estimated coefficients. Other approaches exist, e.g., odds-ratios, that formalize the differences among variables.

Comments for the author

I commend the authors for incorporating suggestions and conducting additional analyses per suggestions. This was no small feat, and as a result the manuscript is greatly improved. I did find it interesting that the new forest analysis did not take advantage of contemporary approaches to multi-model comparison. There are several approaches available, AIC (akaike's information criterion), DIC (Deviance information criterion), and BIC (Bayesian information criterion) being the most common ones in ecology. I recommend using one of these to evaluate a suite of models that includes compactness and forest variables separately and together. Doing so will greatly improve the strength of inference regarding factors affect birds in forested environments.

---

## Author Rebuttal · Round 0.2

Thank you for your submission to PeerJ. I am writing to inform you that in my opinion as the Academic Editor for your article, your manuscript "Forest birds respond to the spatial pattern of exurban development in the Mid-Atlantic region, USA" (#2015:09:6861:0:0:REVIEW) requires a number of major revisions before we could accept it for publication.

The comments supplied by the reviewers on this revision are pasted below. My comments are as follows:

**Editor's comments**
This is an interesting paper and both reviewers saw value in the work. Reviewer 2 outlines some additions to the current analysis, that I think are necessary in terms of knowing how to interpret the results. Below I outline some comments based on my own read of the paper.

Line 123: Although the method used here might reduce the risk of autocorrelation, there could still be spatial structure in the data. Is there justification for subsampling at this scale (e.g., analyses done with these routes, or other work using BBS data)? If not, either testing for autocorrelation or adding it into the models would be helpful.
We tested for autocorrelation using Moran's I and there is no indication of spatial autocorrelation (Moran's I = 0.108, p = 0.182). That information was included in the text (L131).

Line 163: I would specify the pixel size. Many readers will know, but it might be helpful for those unfamiliar with the imagery.
We included in the text that the pixel size of Landsat 5TM used was 30 m (L161).

Line 215: Please specify the R package. (I assume this analysis is not in base R and package titan doesn't look right …)
The source code is available as supporting information in Baker & King 2010 and also an R-package (TITAN2) is available. Thus, we included the name of the package and the reference of the paper (L255).

Line 238: Why give only examples in Fig 3? Since space is not limiting at PeerJ, I see no reason not to provide the results for all species.
We included in Fig 3 all the species for each of the buffer areas as suggested.

Also, perhaps I missed it, but did you directly compare these models to linear alternatives. What evidence is there that the nonlinear models provide better fit to the data?

Besides the visual inspection of the relationships and loess regressions (L198-201), we fitted linear models but the models did not provide a good fit based on the residuals and Normal Q-Q plots (see example below for Eastern Wood-Pewee for the 1 km-radius buffer). If the editor considers necessary we would be happy to include the figures of the 11 species for the two buffers in the paper.

[Figure]

Line 242: I concur with reviewer 2 in my uncertainty about what is meant here. Is it that you mean only 1 of 6 forest indicator species were both significant and reliable (i.e., SCTA, WOTH, etc. are forest indicators)? If so, maybe you could just drop the word "indicator" throughout the paper, since it creates confusion given its variable and often ambiguous use in the literature. If the intended meaning is different, please clarify.
We clarified in the text that only one of the six forest species (i.e. Scarlet Tanager) showed a significant and reliable threshold response to compactness (L269-270). Also, we revised the terminology used and removed the word "indicator" to avoid confusion as also suggested by reviewer 2.

Line 252: By which forest species? There was a range of values for these measures across the forest species.
We added some examples of forest species (e.g., Red-eyed Vireo, Eastern Wood-Pewee) that had similar values to those forest-edge species mentioned and also added a reference to Fig. 4 (L279-280).

Line 258-9: Maybe I am misreading the table, but it looks like 3 of 10 cases involve opposite results at the two scales. Moreover, could the situation for some of the other

cases be unclear because of uncertainty in the estimate of z (I don't see that uncertainty reported); i.e., can you be sure that those cases do not also have a difference that went undetected? For the 3 cases with differences, SCTA and EATO are more reliable at the 400-m scale, while WOTH is more reliable at the 1-km scale. My interpretation does not seem to match what is written in this paragraph. Am I just not understanding Table 2?

We agreed with the editor that the paragraph does not reflect the results presented in Table 2. We modified the paragraph to indicate that in some instances, the direction of the response change with extent of analysis. Wood Thrush responded positively to compactness of exurban development for the 1-km radius buffer. Although the direction of the response changed for the 400-m radius buffer, the indicator was not reliable at this extent (reliability = 0.80). For the other species (i.e. Scarlet Tanager and Eastern Towhee), wide confidence bands and reduced z scores when compared to the reliable extent, highlighted uncertainty when the abundance distributions did not show a clear response. Therefore, where there were differences in the reliability and direction of response at different extents, the 1-km relationships were more reliable (L282-289). In other words, the response is detected when the species is significant and the response is both reliable and pure. When the confidence intervals are wide and the z score decreases the response is uncertain.

Line 274: Were these areas increasingly intact (i.e., through regrowth). If not, I'm not sure how they could account for an increase in forest bird abundance. Simply remaining intact would only account for a lack of decrease.

We clarified in the text that forests adjacent to the study area were protected areas and regrowth forests (L323-324), which provide a potential explanation for abundance increases.

Line 282: The logic behind this argument is unclear to me, because the forest interior species described in this study are not the same species that come into feeders, plantings, etc. and that account for the increase in richness with low level development. I agree with reviewer 2's suggestion that other variables may be confounding the analysis and that forest loss and fragmentation should be considered as covariates.

We agreed with the editor and removed that statement. We focused on forest cover and argue that even though forest decreased around survey stops, forest cover was nonetheless above the minimum amount of habitat necessary for the persistence of forest birds (> 30%; Andrén 1994; Betts et al. 2007; Radford et al. 2005; Suarez-Rubio et al. 2013; Zuckerberg & Porter 2010) (L327-330). As suggested by reviewer 2, we also examined species response to forest loss and fragmentation in relation to compactness of exurban development by performing GAMs. The models used bird species (adjusted bird counts) as the dependent variable and compactness of exurban development, proportion of exurban development, proportion of forest, number of forest patches greater than 0.45 ha, and forest edge as predictor variables. The later variables were estimated following Suarez-Rubio et al. (2013). Gaussian errors and identity link were used and smoothing parameters were automatically selected based on the effective degrees of freedom and a generalized cross validation criterion in R (package: mgcv)

(Wood 2001; Wood 2006). Models were evaluated based on graphical diagnostic plots and the explanatory power of the model was assessed by examining the amount of the explained deviance. Predictors with high significance levels (p < 0.01) were identified as key factors that have strong effects on bird species (L230-247).

Also, the data for the species in Fig 3 suggests that any biological change over the compactness scale has a pretty low magnitude, in relation to the variation in the data. This suggests that the biological effects are very small. Given the other things that potentially affect bird densities, I wonder how biologically important such effects are? Evaluating their importance in light of other things known to affect forest birds would thus seem especially important.

When forest loss and fragmentation were included as predictor variables in the GAMs in addition to the exurban development measures (i.e., proportion and compactness), forest had a highly significant effect on all forest species modeled and most forest-edge species at the 1-km radius buffer (Table 3). Number of forest patches had a significant influence on Red-eyed Vireo and Scarlet Tanager and forest edge did not affect any of the forest species. The effect of exurban development varied among forest species. Only Red-eyed Vireo was significantly influenced by both proportion of exurban development and compactness of exurban development. Eastern Wood-Pewee and Wood Thrush were influenced by compactness of exurban development, whereas Scarlet Tanager was only influenced by proportion of exurban development. None of the forest-edge species were influenced by compactness of exurban development at the 1-km radius buffer, although Eastern Phoebe, Eastern Towhee and Indigo Bunting were affected by its proportion. Regarding forest fragmentation, only Indigo Bunting was influenced by number of forest patches, whereas Eastern Phoebe and Eastern Towhee were affected by forest edge. Neither Gray Catbird nor Northern Cardinal was influenced by forest loss, forest fragmentation, proportion of exurban development and compactness of exurban development. Models at the 400-m buffer and for American Redstart and Ovenbird at the 1-km buffer did not converge. (L296-L314).

Line 293: Indigo bunting declines in the eastern US are also almost certainly attributed to forest regrowth, which has reduced the shrubby habitats that they tend to use. Note that there is a big difference between catbird and indigo buntings. The former are frequently associated with suburbia, whereas the latter are not.

We added that forest regrowth as another reason for population decline of Indigo Bunting and make the distinction that Gray Catbird is frequently associated with suburbia (L345-346).

Fig. 4. Putting species names on both vertical axes seems potentially confusing given that you have two panels side by side. Since you clearly indicate direction of effect with the symbol, I would put them all on the left axis. Alternatively, you could stack the panels on top of each other.

We stacked the panels on top of each other and changed the legend to "direction of effect" as suggested by reviewer 2.

Please be aware that we consider these revisions to be major, and your revised manuscript will probably have to be re-reviewed.

If you are willing to undertake these changes, please submit your revised manuscript (with any rebuttal information*) to the journal within 60 days.

**\* Resubmission checklist:**

When resubmitting, in addition to any revised files (e.g. a clean manuscript version, figures, tables, which you will add to the "Primary Files" upload section), please also provide the following two items:

1. A rebuttal Letter: A single document where you address all the Editor and reviewers' suggestions or requirements, point-by-point.
2. A 'Tracked Changes' version of your manuscript: A document that shows the tracking of the revisions made to the manuscript. You can also choose to simply highlight or mark in bold the changes if you prefer.

Accepted formats for the rebuttal letter and tracked changes document are: DOCX (preferred), DOC, or PDF.

PeerJ does not offer copyediting, so please ensure that your revision is free from errors and that the English language meets our standards: uses clear and unambiguous text, is grammatically correct, and conforms to professional standards of courtesy and expression.

Chris Elphick
Academic Editor for PeerJ
* * *
# Reviewer Comments

## Reviewer 1 (Anonymous)

### Basic reporting

My comments are focused on the use of clear and unambiguous text in the manuscript. A quantitative metric such as compactness and how it changes over time and at different scales does not lend itself to an intuitive interpretation of how, what or why birds might be responding in certain ways within the landscape. Because the language used in the methods section was quite technical, it was challenging to interpret if and how each test was designed to address a potential biological change or a mechanism driving avian response. Although the authors may wish to retain this technical language, statements relating methodological approaches to the biology of how or why birds might be expected to respond to changes in compactness or overall level of development would be helpful. This would also help the reader develop a deeper understanding of the potential mechanisms driving the results, which were contrary to the authors' predicted response.

We revised the language of the methods section to minimize the use of technical terms or clarify what the purpose of the methods was.

In the discussion session, it would be helpful if you could place your results within the larger landscape context. What changes in land use cover are happening at the regional or flyway scale that could be contributing to your results? Are there any limitations or biases associated with BBS survey data that could influence your results that the reader should be aware of?

We added in the discussion that amount of exurban development has expanded substantially in the region and nonetheless the amount of forest habitat is still large to support forest species partly due to adjacent forest regrowth and protected areas (L327-330, 402-405). We also added that there are some limitations and highlighted that the use of bird counts along BBS routes may not reflect occurrence and abundance of more sensitive species such as Kentucky Warbler. Although, counts along roadside have been shown to be representative of changes occurring over a much broad areas (Keller & Scallan 1999) our findings cannot be generalized beyond the range of housing density included in this study (more wilderness areas or more urbanized areas). In addition, the compactness index was developed to assess the clumpiness of exurban housing and assumes that there is development thus it is not suitable to compare with areas without development (L408-415).

Figure 4 – the caption for this figure is quite long and includes information that should be limited to the methods section. Or, possibly there was a formatting error? There are two paragraphs in my version, one that is quite brief, the other quite long.

We revised and shorten the caption of Fig. 4.

**Experimental design**

As relates to the knowledge gap being filled and how this study contributes to filling that gap, I was surprised that the authors failed to mention the large body of work that Dr. John Marzluff and his students and collaborators have produced on the subject of avian responses to urban development, including exurban development. By not referencing this body of work, I'm concerned that the authors might have missed an opportunity to place their research in the context of the current state of knowledge on the subject.

We cited the work of Marzluff and collaborators as suggested by the reviewer (L42, L56-58, L84-86)

**Validity of the findings**

No Comments

**Comments for the author**

This study contributes to a growing body of work documenting avian response to exurban development, identifying threshold responses for species at difference scales of interest. For species that were predicted to decline in response to increased compactness but instead had a positive association, these results suggest that further research on the topic should focus on the specific elements of the landscape that birds are responding to, e.g., increased food availability, human development pressure in surrounding areas, etc., and that it is important to consider multiple scales when

assessing avian response to landscape change.

If the authors are interested in promoting the application of these results in land use planning it would be helpful if they could provide a more straightforward interpretation of how this research might guide the development of more bird-friendly suburbs and exurban areas.
We provided management guidelines at the end of the discussion and emphasized that management efforts should try to concentrate development away from ecological sensitive areas, create or maintain safe zones, and minimize forest loss or forest fragmentation (i.e., increase compactness) to support forest birds (L425-427).

**Reviewer 2 (Chadwick Rittenhouse)**

**Basic reporting**
The article is written well with respect to structure and flow. Some terms require additional or revised explanations to improve clarity, e.g., Compactness index, indicator taxa, indicator response taxa, and indicator response.
We removed the indicator terms and explained what the compactness index was (L202-212). In addition, we included that compactness was defined as the degree of clumpiness of exurban development surrounding each survey stop at each time period considered in the abstract (L24-25).

Line 180-183: Compactness index. I re-read the description several times, and dwelt on figure 2. I'm having a hard time lining up the text with figure 2. The best I can interpret from the figure, the compactness index is:
MSPA all other classes / (MSPA islet + MSPA all other classes)
Which would make low values for compactness when MSPA all other classes is low proportion of the landscape. This is not what the text says. Please clarify.
We thank the reviewer for highlighting this confusing statement. We clarified the text by indicating that the compactness index is a measure of the proportion of exurban development within any MSPA classes other than the islet class (i.e., 1 – (Exurban Development islets / Exurban Development all classes) (L189-191). In addition, we changed the legend of Fig. 2 to make it more intuitive.

Line 242: Term "indicator taxa" – after reading this statement and going back to the description in lines 193 – 202, I'm still lost as to what "indicator taxa" and "indicator response taxa" and "indicator response" means. Coming from the conservation arena, I read indicator taxa and think indicator species, meaning a species whose presence or abundance is representative of certain environmental conditions. Please re-write the explanation of these terms with the goal of moving them from jargon to technical terms.
We revised the description and removed the terms "indicator taxa" and "indicator response taxa" and focused on species and the response that is highlighted by TITAN (L202, 205, 216). Also, we moved away from jargon and explained that the overall goal of TITAN is to distinguish if a species responses to an environmental stressor (in this

case compactness of exurban development) and whether the response is negative (z-) or positive (z+) (L210-212). In the results section, we also amended the text and removed the terms "indicator taxa" and "indicator response taxa" to avoid confusion (L278).

**Experimental design**
The experimental design is sufficient for the question at hand. My comments primarily seek clarification of the methods.
We clarified the methods as suggested by the reviewer.

Line 143 and 146: I see Cit as count for species at stop i at time t, but I don't see where Cit is in the model statement on line 146.
We added in the text that Cit was assumed to be Poisson with mean µit and included the equation (Cit ~Pois(µit) before the full model (L150-151).

Line 156: Was classification accuracy assessed? How? And what are the results?
Classification accuracy was done by randomly selecting 25% of the training data collected from aerial photographs and then calculating overall accuracy using the kappa statistic (Congalton, 1991). Given that a detailed description of the methodology regarding how the exurban development maps were developed is already included in Suarez-Rubio et al. (2012a) and that is not the aim of this study, we only included in the manuscript that the overall classification accuracy for the final exurban development maps ranged from 93 to 98% (kappa: 0.87 to 0.96) (L182-183).

Line 167: Which imagery was classified with MSPA? Was it the (pre-processed) Landsat 5 TM images? The classified Landsat 5 Tm images from the previous paragraph?
We revised the text to clarify that MSPA was used to further analyze pixels belonging to branches of the decision trees that could not discriminate between exurban and urban areas based on spectral characteristics alone (L172-174).

Line 167: I'm familiar with MSPA and have used it in analyses myself. This is a clever application. I see why the edge width of 1 cell (30m pixel) was used – to grab only the isolated cells – and why only certain MSPA classes were considered exurban. Nicely done. A question though, how sensitive is compactness to a change in the edge definition? In the discussion. Lines 313 to 321, the literature suggests edge effects extend much more than 30 m. Where any other edge definitions (e.g., 60 or 90 m) explored?
To create the exurban development maps we used an edge-width of one to be able to extract those isolated cells that could represent scattered isolated housing units. So we used the same edge-width when developing the compactness index to be consistent with the definition that we used. In addition, in the original classification, we compared different edge widths, and the best way to extract scattered isolated pixels resulted from edge-width of one. In addition, when the edge-width was changed to two or three cells for the compactness index, the small core areas that were identified in those isolated

areas disappeared and all cells were classified as islets because if the algorithm does not identify core cells the cells are classified as islets, which did not represent the clumpiness that we were interested on (see figure below for a 1 km-radius buffer example, this was also the case for all sample landscapes). The edge effects that the literature suggests were addressed by evaluating two extents at 400 m and 1 km buffer zones. We are confident that this approach reflects the possible edge effects that exurban development may have instead of structurally increasing the edge-width when exurban is considered as foreground.

[Figure]

Line 202: negative (z-) and positive (z+) indicator response taxa…it's not immediately clear that z is the "effect" of interest and the sign indicates the direction of the effect, e.g., (–) means the species has a negative response to increasing compactness. In Table 2 and Figure 4, is it appropriate to say "Direction of effect" and use the sign, omitting z?
We revised Table 2 and Figure 4 and removed z and replaced it by direction of effect and used only the sign and as suggested by the editor we stacked panel A above panel B.

**Validity of the findings**
My primary concern is in the validity of the findings in light of alternative explanations for the patterns observed here. The authors examined forest and forest-edge bird response to compactness of housing development. The definition of exurban is good, as are the methods to identify it after addressing clarifying comments.
We clarified the methods as previously suggested by the reviewer (see comments above).

However, I'm curious as to how factors known to affect forest birds – forest loss, forest fragmentation, and forest degradation – are treated in the analysis. It seems to me that exurban development may be confounded with forest loss (the houses were built on something, likely forest?) and with forest fragmentation (decrease in patch size, increase in edge without change in forest area). Without accounting for loss or fragmentation in the analysis, an alternative explanation for the results is that minimizing forest loss (increasing compactness) or minimizing forest fragmentation (increasing compactness) are beneficial for forest birds. Please address forest loss and fragmentation in the analysis. This might be done by incorporating the information from Table 1 in the analysis of species responses to compactness.

We agreed with the reviewer that forest loss and fragmentation are key factors that affect forest birds. We followed the suggestion and examined species response to forest loss and fragmentation in relation to compactness of exurban development by performing GAMs. The models used bird species (adjusted bird counts) as the dependent variable and compactness of exurban development, proportion of exurban development, proportion of forest, number of forest patches greater than 0.45 ha, and forest edge as predictor variables. The later variables were estimated following Suarez-Rubio et al. (2013). Gaussian errors and identity link were used and smoothing parameters were automatically selected based on the effective degrees of freedom and a generalized cross validation criterion in R (package: mgcv) (Wood 2001; Wood 2006). Models were evaluated based on graphical diagnostic plots and the explanatory power of the model was assessed by examining the amount of the explained deviance. (L230-247). Given that TITAN only assess one environmental stressor at a time, we used GAMs to identify predictors with high significance levels (p < 0.01) which represented key factors that have strong effects on bird species and to better account for potential non-linear trends between the response and predictor variables.

The introduction should also provide a brief literature review on forest loss vs fragmentation, and the discussion revised to place the results into the broader context of exurban development in forested environments.

We added in the introduction information regarding habitat loss and fragmentation (L56-58, L84-86). In the discussion we added that even though forest decreased around survey stops, forest cover was nonetheless above the minimum amount of habitat necessary for the persistence of forest birds (> 30%; Andrén 1994; Betts et al. 2007; Radford et al. 2005; Suarez-Rubio et al. 2013; Zuckerberg & Porter 2010) (L327-330). We also added that When the effects of compactness of exurban development were assessed in relation to forest loss and fragmentation, proportion of forest had a highly significant effect compared to compactness in most cases (Table 3). This indicates that for forest birds, proportion of forest at the landscape extent may be more important that exurban development. However, proportion of exurban development and compactness had also a significant effect which suggests that if proportion of exurban development or compactness continues this would inevitably lead to the loss of forest species to give a broader context (L331-337).

**Comments for the author**
Overall, this is well-written, well thought-out study addressing a highly relevant question: does compact exurban development reduce negative impacts on forest birds?

The statistical approaches to address this question are top notch, though not complete. Please address how forest loss, forest fragmentation, and forest degradation may be confounding factors in the compactness analysis. This may be accomplished by incorporating the information from Table 1 in the analysis of species responses to compactness.
Please see comments above.

© 2015, PeerJ, Inc. PO Box 614 Corte Madera, CA 94976, USA

---

## Round 0.3 · accepted · Accept

· Academic Editor

Accept

Thank you for your responses. I continue to think that, given the use of a Bayesian framework, the entire analysis could be done in a single model, but that does not mean that the current results are invalid, so I'm happy to accept the current version.